



# Evaluating uncertainty in estimates of soil moisture memory with a reverse ensemble approach

Dave MacLeod[1], Hannah Cloke[2,3], Florian Pappenberger[4,5], and Antje Weisheimer[4,6]

[1]Atmospheric, Oceanic and Planetary Physics, Department of Physics, University of Oxford, Oxford, UK
[2]Department of Geography and Environmental Science, University of Reading, UK
[3]Department of Meteorology, University of Reading, UK
[4]European Centre for Medium-Range Weather Forecasts, Reading, UK
[5]School of Geographical Sciences, Bristol University. Bristol, UK
[6]Department of Physics, National Centre for Atmospheric Science (NCAS), University of Oxford, Oxford, UK

*Correspondence to:* Dave MacLeod (macleod@atm.ox.ac.uk)

**Abstract.** Soil moisture memory is a key component of seasonal predictability. However uncertainty in current memory estimates is not clear and it is not obvious to what extent these are dependent on model uncertainties. To address this question, we perform a global sensitivity analysis of memory to key hydraulic parameters, using an uncoupled version of the land surface model H-TESSEL.

Results show significant dependency of estimates of memory and its uncertainty on these parameters, suggesting that operational seasonal forecasting models using deterministic hydraulic parameter values are likely to display a narrower range of memory than exists in reality. Explicitly incorporating hydraulic parameter uncertainty in models may then give improvements in forecast skill and reliability, as has been shown elsewhere in the literature. Our results also show significant differences with with previous estimates of memory uncertainty, warning against placing too much confidence in a single quantification of
uncertainty.

## 1   Introduction

The persistence of soil moisture is an important aspect of land-atmosphere interactions (Delworth and Manabe (1989); Koster and Suarez (2001); Seneviratne et al. (2006, 2010)). It shows strong coupling of soil moisture with precipitation and temperature (Koster et al. (2004), Fischer et al. (2007),Rahman et al. (2015)), and is a key aspect of weather and climate prediction. The soil
moisture reservoir has a memory considerably longer than most atmospheric processes, and as a low-pass filter lengthens the timescales of climatic anomalies. Persistence in soil moisture is linked to persistence in humidity, temperature and precipitation (Delworth and Manaba (1993); Koster and Suarez (1995); Pal and Eltahir (2001), amongst others). It prolongs the effect of drought (Nicholson (2000)), enhances the severity and persistence of floods (Bonan and Stillwell-Soller (1998); Hong and Kalnay (2000); Liu et al. (2014)), impacts the length of heatwaves (Lorenz et al. (2010)) and can determine the predictability
of atmospheric surface climate anomalies (Wang and Kumar (1998); Douville (2004)).





Previous work has analysed soil moisture memory in uninitialised atmospheric global circulation models (Wu and Dickinson (2004); Seneviratne and Koster (2012)), showing significant regional differences in memory. Other work has looked at estimating soil moisture memory from observations of streamflow (Orth et al. (2013)). It has also been demonstrated that the choice of soil hydrology scheme impacts simulated soil moisture memory (Hagemann and Stacke (2014)).

5 We take an alternative approach here, inspired by previous work on hydrological prediction uncertainty (Wood and Lettenmaier (2008)). We run a hindcast in a setup analogous to a standard seasonal hindcast setup, using an uncoupled model with ensembles of initial states and forcing time series. This allows us to compare memory estimates of a seasonal forecast system when forecasts are initialised at different points in the year, as well as looking at spatial differences. It also allows us to make an estimate of the sensitivity of estimates of soil moisture memory to uncertain hydraulic parameters.

10 This approach may be compared to work at longer decadal timescales following a similar approach (Corti et al. (2015)). By using different combinations of initial conditions and forcing years to run global climate models, they analyse the crossover point when forcing becomes more important than initial conditions and so identify the relative importance of initial conditions and forcing.

In concert with this work, our work here aims to understand the importance of initialization of slowly moving components 15 of the climate system for longer term climate prediction. In addition, this approach allows new insight into the controls that hydraulic parameters have on the generation of soil moisture memory and the relative uncertainty compared to initial conditions.

The model, data and experimental methods are described in the following section, after which results are described. The final section of the paper contains a discussion of results and conclusions.

## 2 Methodology

20 Wood & Lettenmaier define a method known as reverse-Ensemble Streamflow Prediction (reverse-ESP, Wood and Lettenmaier (2008)). Rather than standard ESP, where one initial state is forced by an ensemble of forcings, this method takes the opposite approach and forces an ensemble of initial states with a single forcing timeseries. In their work they analyse the relative importance of initial conditions and forcing by comparing the spread of ESP and reverse-ESP ensembles.

Inspired by this method, we take a reverse-ESP approach to evaluate soil moisture memory. By forcing an ensemble of 25 initial conditions with the same forcing timeseries, soil moisture states converge, at which point one can say that the soil has lost memory of initial conditions. This is demonstrated with some example data in figure 1. At the initial point the standard deviation of the ensemble is large and over time reduces at differing rates dependent on forcing conditions, precipitation, evaporative demand and model parameters. By calculating the lead time at which the spread of the ensemble is reduced significantly from its initial spread we can estimate soil moisture memory. This approach enables us to perform a sensitivity 30 analysis of this soil moisture memory estimate to model hydraulic parameters.





## 2.1 Land surface model, forcing and initial data

The land surface model used is H-TESSEL, the Tiled ECMWF Scheme for Surface Exchanges over Land (TESSEL) with revised land-surface hydrology (Balsamo et al. (2009)). This comprises a surface tiling scheme and a vertically discretized soil, with soil layers below ground at 7, 21, 72 and 189cm. Soil moisture is defined for a layer as the volume of water contained in the layer expressed as a fraction of the layer volume. H-TESSEL includes the van Genuchten formulation for hydraulic conductivity (van Genuchten (1980)) and a spatially varying soil-type map.

Running the model in uncoupled mode allows a much larger number of experiments to be run compared to coupling with an interactive atmosphere. This requires forcing at each timestep. To provide this forcing, we use the WFDEI meteorological forcing data set, data at $0.5°$ spatial resolution, created by using the Watch Forcing Data methodology applied to the ERA-Interim data (Weedon et al. (2014)). The forcing comprises of three hourly data of the following variables: longwave & shortwave radiation, rainfall, snowfall, surface pressure, wind speed, air temperature and humidity. Data from the 25 years 1981-2005 inclusive is used, in each case using the four-month period from an initial date to force the model, mimicking seasonal hindcasts (Weisheimer et al. (2014)).

For initial land surface conditions of soil temperature, moisture and ice temperature where frozen soil is present, we use the ERA-Interim Land reanalysis (Balsamo et al. (2015)), using initial states from 1st May and 1st November for every year 1981-2005 inclusive. For climatological data for the model (e.g. albedo, vegetation cover, soil type) we use the H-TESSEL default values.

## 2.2 Hydraulic parameter perturbation

We perturb the van Genuchten $\alpha$ parameter and the saturated hydraulic conductivity. These parameters are related to the movement of water through the soil and their default values in the model setup are based on the FAO soil map of the world (see Balsamo et al. (2009) for details). In reality there is a large uncertainty in these parameters; observed standard deviation in their values across soil samples can be as much as twice the mean (see MacLeod et al. (2015); Carsel and Parrish (1988) for further details). Previous work has shown that these parameters are particularly sensitive (Cloke et al. (2011)).

Here we perturb these parameters by picking a value for each simulation for each parameter from the five-member set $\{-80, 40, 0, +40, +80\}\%$ and applying this perturbation to the default value for each gridpoint. Perturbing both parameters in this way gives a 25 member ensemble. Though these perturbations are relatively large and there is a weak correlation between these two parameters, the perturbed range is within the range of observed variability in parameter values. Furthermore the parameters which are used have to be considered as effective parameters as measurement scale and model scale differs significantly (Barrios and Francés (2012)). This leads to significant uncertainty when attempting to define parameters on the model grid scale.

A simulation is run for every combination of 25 initial states, 25 forcing states and 25 parameter perturbations, giving a total of $25^3 = 15,625$ runs. Due to computing limitations we run at a reduced spatial resolution with $18 \times 36$ gridpoints globally (note that these points are 'picked' from the original resolution of the forcing and initial conditions instead of being interpolated





over a whole gridbox). Note one caveat is that initial conditions are taken from reanalysis which is created using HTESSEL with default parameters. The land surface temperature and moisture climatology is slightly influenced by these parameters, however to recreate a new set of reanalysis for each parameter combination we investigate here is far beyond the scope of the study.

## 2.3 Experimental setup

For the ensemble of initial condition data we run an ensemble of simulations using the same year's forcing for each ensemble member, using a constant parameter perturbation throughout each simulation. An example of this is shown in figure 1. We characterise the rate of memory loss by defining the date of memory loss $t_{ml}^{fp}$, the time from initialization when the spread of soil moisture in the ensemble (forced by year $f$ and parameter combination $p$) has reduced to $1/e$ of its initial value, that is:

$$t_{ml}^{fp} = \sup\{t : \sigma(t) \leq \sigma(0)/e\}, \tag{1}$$

where $\sigma(t)$ is the standard deviation of the ensemble at time $t$, $\sigma(0)$ is the initial spread and the memory loss time $t_{ml}$ is defined by the supremum norm (i.e. the maximum value of $t$ bounded by $\sigma(t) \leq \sigma(0)/e$. $t_{ml}$). We calculate $t_{ml}^{fp}$ for each forcing year, and take the average across all 25 forcing years:

$$\langle t_{ml} \rangle^p = \sum_{f=1}^{25} t_{ml}^{fp}, \tag{2}$$

to give a more robust estimate of the date of memory loss, for each spatial gridpoint and parameter combination. We can then calculate $\langle t_{ml} \rangle^p$ separately for each parameter combination, giving a measure of the regions where variations in hydraulic parameters give the largest changes in memory loss. This can be expressed by the standard deviation of $\langle t_{ml} \rangle^p$ across the parameter ensemble:

$$\sigma_{t_{ml}} = \sqrt{\frac{\sum_p^{25} (\langle t_{ml} \rangle^p - \langle t_{ml} \rangle)}{25}}, \tag{3}$$

where $\sigma_{t_{ml}}$ is the standard deviation across the perturbed parameter ensemble, $\langle t_{ml} \rangle^p$ is the memory loss date for parameter combination $p$, and $\langle t_{ml} \rangle$ is the average date across all combinations. Small variations in memory are less meaningful when memory is large (and more meaningful when memory is normally small) so to highlight these more meaningful variations to hydraulic parameters we re-express the standard deviation as the coefficient of variability, i.e. as fraction of the mean time of memory loss, expressed as a percentage:

$$COV = \frac{\sigma_{t_{ml}}}{\langle t_{ml} \rangle} \times 100\%. \tag{4}$$





## 3 Results

### 3.1 Estimation of soil moisture memory

Figure 2 shows the average date of memory loss for the default model parameters ($\langle t_{ml}\rangle^p$, where $p$ is the default parameter set) for May and November start dates, for the top vertical soil level. Results for level two and three are discussed below

but presented in the supplementary material. We do not discuss the fourth soil level as the free drainage condition impacts negatively on the realism of the exact values of soil moisture here.

In general memory increases with depth, though with significant differences spatially and between start dates. The longest memory is seen in regions in northeastern Asia and the extreme northwest of the Northern American continent. For May several points have a longer memory of between 20-60 days, and in the furthest north one point still retains memory of the initial state

at the end of the simulations (120 days). Similarly the longest memory in November is seen for the same regions - however the memory is much longer in these regions for this start date, and the region of long memory extends much further south, down the Rocky Mountains in the US, and the Himalayas in Asia. This long memory is likely due at least in part to snow cover, when soil is insulated from any precipitation forcing, allowing persistence of an initial soil moisture state (Koster et al. (2010)).

Correspondingly, the surface layer in the Southern Hemisphere has a relatively short memory for both start dates, likely

in part due to an absence of snow cover on Southern Hemisphere land (excluding Antarctica). The second and third levels generally have longer memory than the surface, as the influence of precipitation forcing on soil moisture is damped. This is partly due to the loss of some moisture by evapotranspiration before reaching the lowest levels.

### 3.2 Sensitivity of estimation of soil moisture memory to hydraulic parameters

#### 3.2.1 Variation in memory loss date

Figure 3 shows the standard deviation in the average date of memory loss across the parameter ensemble ($\sigma_{t_{ml}}$) for May and November start dates for the top level. The absolute variation in memory length with parameter does not follow exactly the same pattern as the magnitude of memory. For May the largest variations in memory are seen in snow-covered regions in northeast Asia, but also in southern Africa. Similarly, November shows large variation in memory over the gridpoints around the Himalayas, yet much smaller variation around the areas of Asia and the US with long memory. The largest variation in

memory for November in fact occurs in Europe and western Asia, where memory is short.

In general for both start dates lowest absolute variation in memory length is seen at the top level, with an increase in variation with depth (lower levels shown in supplementary material). The highest absolute variation in memory length is seen at level three, with a standard deviation in the date of over a month. In May this is spread over Eurasia and North America, though with some high variability also seen in the Southern Hemisphere. The top layer shows a particular sensitivity to parameters in

the Northeast Asia region where memory is highest.

November start dates also show the same pattern of more variability at lower levels where memory tends to be longer, however the spatial pattern tends to be more heterogeneous than May start dates. For instance, variability in May in the third



level is roughly constant over much of Asia, whilst in November there is a sharp contrast, with a band stretching from Northeast to Southwest where variability in the date is quite low. This heterogeneity extends upwards to higher levels, with the highest variations in November contained mostly in the Western half of Eurasia. There is also a clear distinction between Western and Eastern North America in the third level for November starts, with much higher sensitivity in the East than the West. This however does not extend up to levels one and two, where the sensitivity is uniform across the continent.

### 3.2.2 Coefficient of variation of memory loss date

Looking at variation of memory with parameters may be in some cases somewhat dependent on the original magnitude of memory. To understand where the memory is particularly sensitive we show the coefficient of variation of memory loss date(figure 4).

This highlights memory estimates in Europe as being particularly sensitive to the uncertainty in hydraulic parameters for November-initialized forecasts, with standard deviation in the memory loss date as over 50% of the mean at the surface, whilst points in Asia are less than 20%. Similarly, Eastern North America and Northern South America have high sensitivity, whilst the West and South respectively do not. Sensitive regions in November also include Central Africa, Indonesia and some points in the Middle East, with a slight reduction in magnitude and modification of the spatial pattern for lower levels.

For the May start date the spatial pattern is quite different from November, with low sensitivity in Europe and Eastern North America. Sensitive regions are found in Southern Africa, West North America and Northeast Asia. The spatial pattern of sensitivity changes quite significantly with depth in May, with high sensitivity at the third layer for Western North America, for a band running across North Asia and South America.

## 4   Discussion and Conclusions

Here we have developed a method to calculate soil moisture memory, and used this to estimate its sensitivity to hydraulic parameters. These hydraulic parameters influence the speed of the movement of water through the soil, so it is not surprising that the persistence of initial moisture states is influenced by them. However soil moisture memory emerges from the interplay of precipitation, evaporative demand and drainage conditions over time, and the pattern of sensitivity is not clear *a priori*. The current study highlights regions for which memory is particularly sensitive to hydraulic parameters and also indicates differences in memory and sensitivity between hindcasts initialized in different seasons.

Previous work has attempted to characterise soil moisture memory, most extensively using the GLACE AGCMS (Seneviratne et al. (2006)). This work analysed the output of 12 AGCMs over the period 1 June to 31 August, forced by SSTs from 1994. Roughly corresponding to the same period as the May start used here, this work looked at geographical variations in average memory and in the standard deviation of memory estimates across the model ensemble. Their findings are not in full agreement with our work here; amongst other things, our results also do not lead to their conclusion that memory is higher in the midlatitudes than the tropics. Furthermore the highest uncertainty in the GLACE AGCMs is found in regions of low soil moisture memory. Our work shows deviation from this result, with large uncertainty in both regions of high and low memory.





These discrepancies are likely to have several explanations. A key reasons for differences in uncertainty estimates is that the focus of the uncertainty quantified is different; Seneviratne et al. (2006) look at a multi-model ensemble, which gives some exploration of model and structural uncertainty, whilst in our study we use a single model and look at the uncertainty in hydraulic parameters.

Note that we do not claim that our own sensitivity analysis is total, only that we show how memory estimates using a single model are sensitive to two specific model hydraulic parameters. It is likely the case that there remain unexplored uncertainties in the land surface, for instance: rooting depth, the depth of the free drainage lower boundary condition and moisture thresholds (such as permanent wilting point, field capacity and saturation). Furthermore other models may show more or less sensitivity to the hydraulic parameters we test here. However the main focus of this work is to demonstrate the uncertainty in estimating soil

moisture memory, and the potential importance of hydraulic parameters. This is relevant for operational seasonal prediction, in which land surface models are run without any variation in these uncertain parameter values. Given the key role in memory in high impact events such as flooding, droughts and heatwaves, a forecasting system with deterministic parameters is likely to display a narrower range of possible memory than exists in reality. This is likely to give negative impacts on forecast skill and reliability, or put more optimistically: by explicitly including parameter uncertainty in operational probabilistic forecasts (which

already incorporate initial condition uncertainties) we may improve forecast skill and reliability. Indeed, coupled seasonal forecasts which include an explicit representation of hydraulic parameter uncertainty have already shown improved prediction of the 2003 European heatwave (MacLeod et al. (2015)).

    The results presented here also highlight a more general conclusion that cannot be overstated: that of the uncertain nature of uncertainty quantification. It is almost impossible to explore the full space of our ignorance in models; a multi-model

approach neglects uncertain parameters, whilst a perturbed parameter approach ignores structural uncertainty (and indeed, other parameters). Conclusions about relative levels of sensitivity and uncertainty must therefore be drawn with caution, bearing in mind the limitations of the experimental method.

*Author contributions.* DM conceived the study, ran simulations and wrote the first manuscript draft. Interpretation of results, development of analysis and paper revision was a collaboration between all co-authors.

*Acknowledgements.* DM would like to thank Gianpaolo Balsamo, Emanuel Dutra and Filip Vana at ECMWF for help setting up the model and accessing data. DM and AW would like to acknowledge funding from the EU-FP7 project SPECS (grant agreement 308378). HC would like to acknowledge funding from the NERC project IMPETUS (grant agreement NE/L010488/1). FP would like to acknowledge funding from the H2020 project IMPREX (grant agreement 641811).



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







**Figure 1.** An example of the evolution of top level soil moisture from an ensemble of 25 initial states, each forced with the same year's forcing (in this case, 1981). Data from each plot is taken from different gridpoints, chosen to demonstrate a situation of long and short memory . Note that in the latter case memory of initial conditions is lost by around day 20, whilst the former case retains knowledge of the initial state for several months.





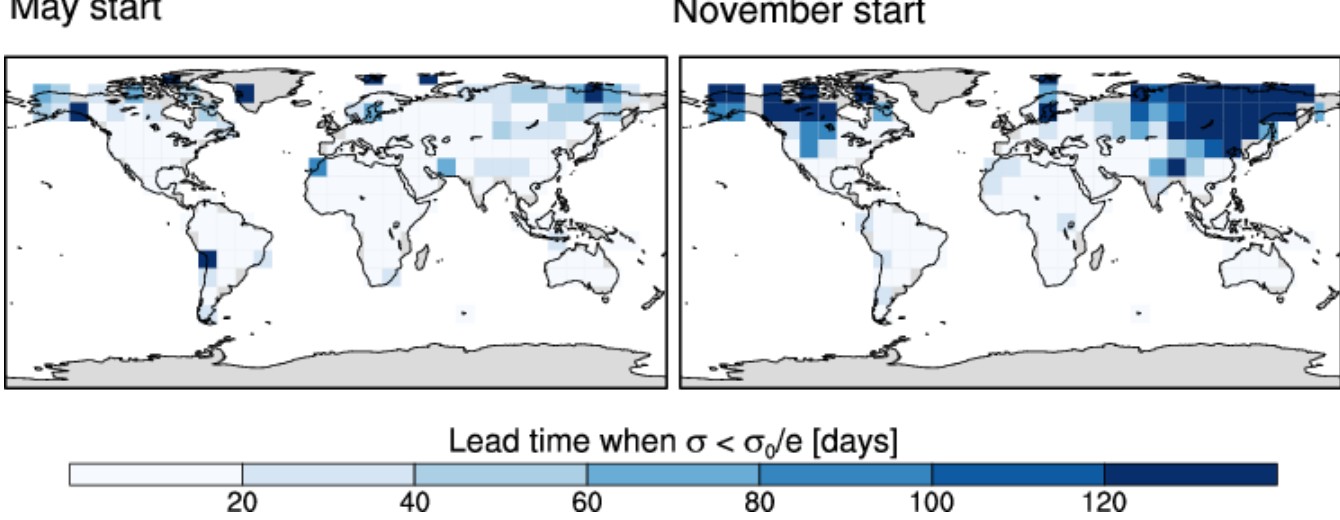

**Figure 2.** Average date of memory loss in the top model soil level, estimated from HTESSEL (default parameter set), for May (left) and November (right) start dates.

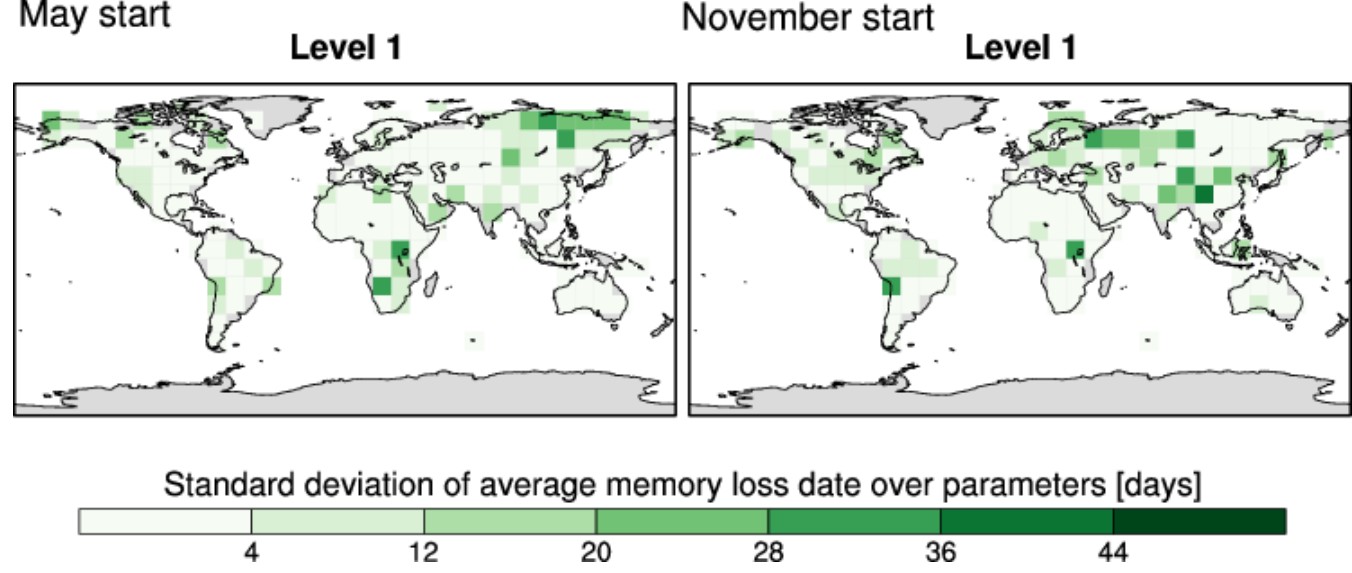

**Figure 3.** Standard deviation in the top model soil level, estimated from HTESSEL, for May (left) and November (right) start dates.





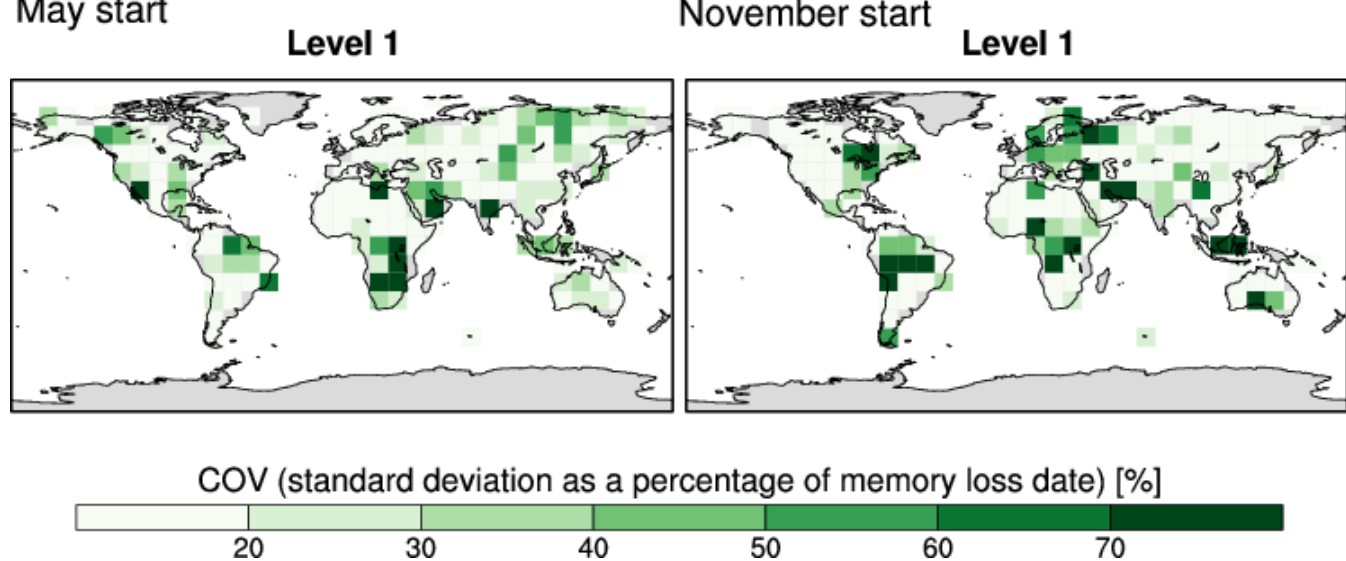

**Figure 4.** Sensitivity of memory to hydraulic parameters (standard deviation of memory loss across hydraulic parameters as a percentage of the memory loss) in the top model soil level, estimated from HTESSEL, for, May (left) and November (right) start dates.