# Peer review of "Evaluating uncertainty in estimates of soil moisture memory with a reverse ensemble approach"

_Hydrology and Earth System Sciences, 2016_

## Referee Comment (RC1) · R. Orth (Referee) · 10 Mar 2016

R. Orth (Referee)

rene.orth@env.ethz.ch

Review of MacLeod et al. 'Evaluating uncertainty in estimates of soil moisture memory with a reverse ensemble approach'

This paper illustrates the importance of particular hydraulic model parameters for the resulting simulated soil moisture memory. Large memory changes are found when varying these parameters within the range of the respective observed values.

————————————

General comments:

The paper addresses a relevant topic and should be of interest for HESS readers. Soil moisture memory has been shown to be of importance for weather and climate

forecasting. It is known that memory estimates vary considerably across models, which might be related to the fact that there are only few available soil moisture observations.

I think the present study is useful in this context to better understand potential sources of uncertainty in simulated soil moisture persistence.

However, I think the analysis needs to be extended. The authors should show why and how the variations in soil moisture memory matter, i.e. for example how they impact the models' weather forecasts. One way to do this might be to link the simulated soil moisture memory and its variations with for instance temperature forecast skills. Such an analysis might reveal potential to improve the use of soil moisture information in HTESSEL. In Orth and Seneviratne (2014) we found for example that at 4 weeks lead time better temperature forecasts can be derived from soil moisture (from another product) alone compared with the respective skill of the ECMWF system.

Furthermore you need to better motivate your analyses of the hindcasts initialized in November. As you mention, the memory results are dependent on snow cover. If the snow isolates the soil hydrology from the atmosphere and prevents any soil moisture-atmosphere interactions, then why should we worry about uncertainty in soil moisture memory in winter?

In view of the outlined problems, I recommend major revisions.

I do not wish to remain anonymous - Rene Orth.

————————————

Specific comments:

page 1 line 1: not clear what predictability you mean here line 9: double 'with' line 13: what does 'It' refer to? line 15: insert 'it' after low-pass filter line 17: soil moisture memory is linked to persistence in temperature and precipitation through its link with evapotranspiration persistence (Orth and Seneviratne 2013)

page 2 line 9: double use of 'estimate' line 27: add 'such as' after 'forcing conditions'

page 3 section 2.1: discuss implications of uncoupled vs. coupled setting for soil moisture memory and its potential impact on surface weather line 10: 'comprises of three', please correct line 16: albedo and soil type are not exactly 'climatological data' lines 20-24: give more reasoning on why these particular parameters were chosen line 28: correct 'differs'

page 4 equation 2: this is a sum, not an average lines 21-23: I see what you mean, but please rephrase to clarify your point here

page 5 line 3: 'p' is used already on the previous page

page 6 line 8: please find a clearer expression for 'memory loss date' line 8: insert space after 'loss date' line 26: correct 'AGCMS'

page 7 lines 14-17: Orth et al. (2016) also shows how weather forecasts can be improved by accounting for land surface model parameter uncertainty line 18: remove 'that of' lines 18-22: either here or elsewhere in the paper, mention that despite the uncertainty found in soil moisture memory its overall magnitude in HTESSEL is comparable with estimates from soil moisture observations (Orth and Seneviratne 2012)

References:

Orth, R., E. Dutra, and F. Pappenberger, 2016 Improving weather predictability by including land-surface model parameter uncertainty. Mon. Weather Rev., doi: 10.1175/MWR-D-15-0283.1, in press

Orth, R., and S.I. Seneviratne, 2014 Using soil moisture forecasts for sub-seasonal summer temperature predictions in Europe. Climate Dynamics, 43 (12), 3403-3418, doi: 10.1007/s00382-014-2112-x

Orth, R., and S.I. Seneviratne, 2013 Propagation of soil moisture memory to streamflow and evapotranspiration in Europe. Hydr. Earth Syst. Sci., 17, 3895-3911,

doi:10.5194/hess-17-3895-2013

Orth, R., and S.I. Seneviratne, 2012 Analysis of soil moisture memory from observations in Europe.   J. Geophys.   Res.   - Atmospheres, 117, D15115, doi:10.1029/2011JD017366

---

## Referee Comment (RC2) · J. Danhelka (Referee) · 12 Mar 2016

During recent years, several studies demonstrated how seasonal hydrological predictability differs across regions and time of the year. This paper applies reverse-ESP approach in order to research the impact of change of parameters of hydrological model.

Generally, this study contributes to the understanding of seasonal predictability and it highlights the importance of "hydrological parameter uncertainty" in this area. As such, I consider it relevant for publication in HESS. The limit of the study is a sparse resolution of application given by the high computational demand of selected ensemble approach. From that point of view, reduced resolution is understandable, but more explanation of selection of grid points is needed, as well as this feature of the study and its impact

on results should be properly discussed. In addition, results are presented only from the point of geographic location of sensitivity anomalies, however if grid characteristics were "picked" from higher resolution data, a link to general orography and vegetation type should be given (e.g. an obvious anomaly of average day of memory loss for level 2 and 3 in North African and Middle East deserts.

Secondly, study uses two initial dates (May and November) and 4 months "forecast period", which however will be for given months heavily impacted by snow occurrence and melting. Different starting dates, even with relatively small temporal shift (e.g. by 1 month) would likely provide different results. I am aware that selection of additional starting dates would be computationally extensive and full year coverage was outside the scope of the study, but I feel that some discussion of this issue would be beneficial for readers.

I also advice authors to try to more explicitly describe (discriminate) the implications of their findings from the perspective of atmosphere-land coupled model on one hand, and from the perspective of seasonal flow forecasting applications.

In conclusion, the paper should be subject to some major revisions reflecting above mentioned comments.

A few typing errors appear in the text, among others: Page 1, line 9 – repetition of "with" Page 3, line 10 - "The forcing comprises three hourly data..."

---

## Author Comment (AC1) · 5 Apr 2016

We thank the reviewer for their insightful comments. The first comment made is that the analysis should be extended to show how variations in soil moisture memory matter, for example how they impact models' weather forecast.

This is a good point and we agree in principle. However the analysis we have done is based on an uncoupled land-surface model, and to look at the impact on the output of weather forecasts would mean running the land surface coupled to an atmospheric model, which is not a trivial task for two reasons. One; the setup of the experiment is not a simple job and two; the number of simulations carried out in the same setup (15,625 per start date) would be prohibitive, both in terms of computer time and data storage.

[Figure]

We are certainly interested in the impact of memory on weather forecasts, however in addition to the reasons listed above we believe that the experimental design would be qualitatively different enough from the current analysis to render the investigation out of scope of the current focus. We are however intending to pursue this idea in future and are open to collaboration and would welcome further discussion with the reviewer if they are interested in this.

The second point raised by the reviewer refers to the November initialized simulations, suggesting we should better motivate analysis of hindcasts initialized in November and raising the question about why we should worry about uncertainty in soil moisture memory in winter if snow isolates soil hydrology from the atmosphere and prevents soil moisture-atmosphere interactions.

Our response is twofold. Firstly we use November forecasts as we wanted to focus on the two standard contrasting seasons, boreal summer and winter and many regions for November initializations are unaffected by snow. Secondly, whilst it is true that the snow will insulate the soil from the atmosphere, over the course of the season snow can melt and ultimately restore interaction between the land surface and the atmosphere. At this point initial soil moisture memory anomalies can influence the atmosphere, thus the variations in memory can still be important despite an absence of soil moisture-atmosphere interaction for part of the season. However it is true, that smaller variations in memory are less important when memory itself is long, as shown in the COV plot for the November start date in figure 4.

Naturally we plan to address the main reviewer comments in any future versions of the manuscript, along with including the minor revisions suggested by the reviewer.

---

## Author Comment (AC2) · 5 Apr 2016

We thank the reviewer for the useful comments. The reviewer makes three main points which we will discuss in turn.

The first point suggests we make more discussion of the explanation of why we chose certain gridpoints, and suggested giving more information about general orography and vegetation.

We chose the grid points indiscriminately, in that we took every 10th point from the underlying data without considering a priori the local characteristics. To use low resolution data is suboptimal, since regions of high spatial heterogeneity or complex topography may show memory characteristics which are somewhat dependent on choice of point. However the expense of rerunning the experiment at higher resolution puts this option

outside the scope of this study.

Though our results are unable to capture finer scale variations in moisture memory and its uncertainty, they do give a broad picture of the global pattern. Futhermore, where there is spatial coherence across several gridpoints this gives confidence that the results do not arise from choosing a certain point, but are indicative of a larger patterns across a region (e.g. in figure 4 Southern Africa and Northeast Asia for the May start date and East North America, North South America, Europe and Equatorial Africa for the November start date).

In terms of giving more information about orography and vegetation type, this is a good idea, and in a future version of the manuscript we will provide this as supplementary information.

The second point raises the issue of sensitivity to start dates (whilst acknowledging the computational expense) and suggests more discussion.

We agree, and ideally we would look at multiple start dates. However as the reviewer points out, repeating for extra extra start dates is computationally prohibitive. We plan to add some discussion on this point on the manuscript, namely that whilst boreal winter is impacted by snow in the Northern Hemisphere, results for the Southern Hemisphere in the same season are unaffected. Furthermore the November start date result in Northeast Asia does give some indication of what results look like where snow is present.

The final point asks to try to more explicitly discriminate implications of findings from the perspective of atmosphere-land coupled model on the one hand and from the perspective of seasonal flow forecasting applications on the other.

This is a good comment and important to make this distinction. We would suggest that variations in memory and its uncertainty are important from the perspective of atmosphere-land coupled modelling since the feedback between the land and the at-

mosphere mean that anomalies in soil moisture which persist over a long time can influence atmospheric conditions over an extended period into the future.

For example, a persistent dry soil may exacerbate heating anomalies such as in the case of the 2003 European heatwave. A long memory means that the land surface becomes one of several sources of predictability for the atmosphere. This is only true for situations where there is strong soil moisture-atmosphere coupling, where variations in soil moisture strongly influence the atmosphere via latent and sensible heat fluxes. In regions where coupling is absent, soil moisture memory is less important.

In contrast soil moisture memory is important for hydrological applications independently of the strength of land-atmosphere coupling. Soil moisture is one of the main controls on both rainfall runoff generation and evaporation. Uncertainty and variability in the persistence in soil moisture can then directly influence the predictability of streamflow, with implications for effective flood forecasting and water resource management.